# Paleobiological implications of chevron pathology in the sauropodomorph *Plateosaurus trossingensis* from the Upper Triassic of SW Germany

Joep Schaeffer[1,2]*, Ewan Wolff[3], Florian Witzmann[4], Gabriel S. Ferreira[5,6], Rainer R. Schoch[1,2], Eudald Mujal[1,7]

1 Staatliches Museum für Naturkunde Stuttgart, Stuttgart, Germany, 2 Institut für Biologie, Universität Hohenheim, Stuttgart, Germany, 3 Honors College, University of New Mexico, Albuquerque, NM, United States of America, 4 Museum für Naturkunde Berlin, Leibniz Institute for Evolution and Biodiversity Science, Berlin, Germany, 5 Senckenberg Centre for Human Evolution and Palaeoenvironment (HEP) at the University of Tübingen, Tübingen, Germany, 6 Department of Geosciences, University of Tübingen, Tübingen, Germany, 7 Institut Català de Paleontologia Miquel Crusafont (ICP-CERCA), Universitat Autònoma de Barcelona, Catalonia, Spain

* joep.schaeffer@smns-bw.de

**Data Availability Statement:** All 3D models and CT scans are available on www.figshare.com using the DOI 10.6084/m9.figshare.26044132.

## Abstract

Paleopathology, the study of diseases and injuries from the fossil record, allows for a unique view into the life of prehistoric animals. Pathologies have nowadays been described in nearly all groups of fossil vertebrates, especially dinosaurs. Despite the large number of skeletons, pathologies had never been reported in the sauropodomorph *Plateosaurus trossingensis*. Here we describe the first pathologies of *Plateosaurus* using two individuals with pathologies in the chevrons of the tail, from the Upper Triassic of Trossingen, SW Germany. The two specimens each contain three consecutive pathological chevrons. Our results show that the pathologies were caused by external trauma in one individual and potentially tendinous trauma in the other. Healing of the lesions allowed survival of both animals. Using additional pathological specimens found in other collections and from multiple localities, we observe that 14.8% of all individuals of *Plateosaurus* contain pathologies within their chevrons, suggesting it was a vulnerable bone.

## 1. Introduction

Paleopathology is the study of illnesses and injuries from the fossil record. It is a very important means of understanding the evolution of pathogens, immune systems, intra- and interspecific interactions, and the interaction with the environment animals lived in [1–5]. During the past decades of research, pathologies have been identified in nearly all clades of dinosaurs and many other vertebrates [6–12]. Those pathologies range from trauma to cancer, infections, periostitis, arthritis, and gout among others [4, 9, 10, 13–15]. Most of the dinosaur pathologies

**Funding:** The author(s) received no specific funding for this work.

**Competing interests:** The authors have declared that no competing interests exist.

are described from theropods and ornithischians, however a large number of pathologies in sauropods are also known (e.g. [3, 5, 6, 8, 16–23]).

Several pathologies have already been described from non-sauropod sauropodomorphs. Butler et al. [24] described a specimen of *Massospondylus carinatus*, a common taxon from the Lower Jurassic of South Africa, with the distal end of the tail missing and a lot of reactive overgrown bone around caudal vertebrae 23 to 25, fusing them together. These authors suggested that the distal end of the tail was potentially amputated by a predator, with the *Massospondylus* surviving the attack. Xing et al. [8] presented a specimen of *Lufengosaurus huenei*, from the Lower Jurassic of China, which had its 7th and 8th cervical vertebrae fused by reactive bone. Another pathology in *L. huenei* was described by Xing et al. [9], which consists of a hole in the 3rd right dorsal rib, possibly caused by a predator attack. Using μCT-imaging the authors found evidence of osteomyelitis in the healing bone. Lastly, Cerda et al. [19] observed pathological bone growth in the medullary cavity of *Mussaurus patagonicus*, from the Upper Triassic of Patagonia, Argentina.

*Plateosaurus trossingensis* is one of the most common dinosaurs from the Upper Triassic of Europe and one of the best known non-sauropod sauropodomorphs in the world [25]. Nearly 200 individuals have been found over the past 180 years comprising many relatively complete and partial skeletons and isolated bones [26–30]. Within such a large sample size it is almost certain that several pathologies are present. After inspecting approximately 100 individuals from various localities and collections, several individuals with bone pathologies were identified. Of those ca. 100 individuals, 27 preserved a significant series of chevrons (i.e., haemal arches), and four of those 27 individuals had pathologies on their chevrons. The tail of *P. trossingensis* consists of approximately 50 vertebrae with accompanying chevrons. The chevrons of *P. trossingensis* are generally straight and craniocaudally narrow proximally. More caudal in the tail, they become caudally curved and craniocaudally wider in proportion to the dorsoventral length (Fig 1A).

In this work we aim to provide insight in the paleoecology of *Plateosaurus* by analyzing pathological specimens. Specifically, we focus on two individuals of *P. trossingensis* from the Upper Triassic (Norian) of Trossingen, Germany [30] with three consecutive chevrons displaying pathologies (Fig 1B and 1C). By describing the external and internal morphology we aim to understand the processes forming these pathologies from etiopathogenic and paleobiological perspectives. Furthermore we survey all available relatively complete specimens of *Plateosaurus* from multiple Upper Triassic localities for the broader biological and ecological causes and effects of these pathologies.

## 2. Geological setting

The two specimens herein studied come from the Obere Mühle locality in Trossingen, SW Germany. It is the type locality of *Plateosaurus trossingensis*, and one of the richest dinosaur fossil sites worldwide, with more than 80 relatively complete skeletons and other dozens of partially articulated skeletons and many isolated bones [25, 27, 29–34]. The outcropping deposits mainly consist of a siliciclastic mudstone succession ~12 m thick that belongs to the Trossingen Formation, also known as *Knollenmergel*, of Norian (Late Triassic) age [30, 35, 36].

The sedimentary succession in the locality consists of overall purple- and brown-colored siliciclastic mudstones (classically a so-called red-bed succession) and occasional grey/green-mudstones and marlstones of limited lateral extension interbedded. As a whole, the succession appears relatively monotonous, mainly distinguished by successive events of pedogenesis represented by brecciated horizons and carbonate nodules vertically arranged; mud-cracks are visible in some levels. All in all, this indicates that the original clay deposits underwent

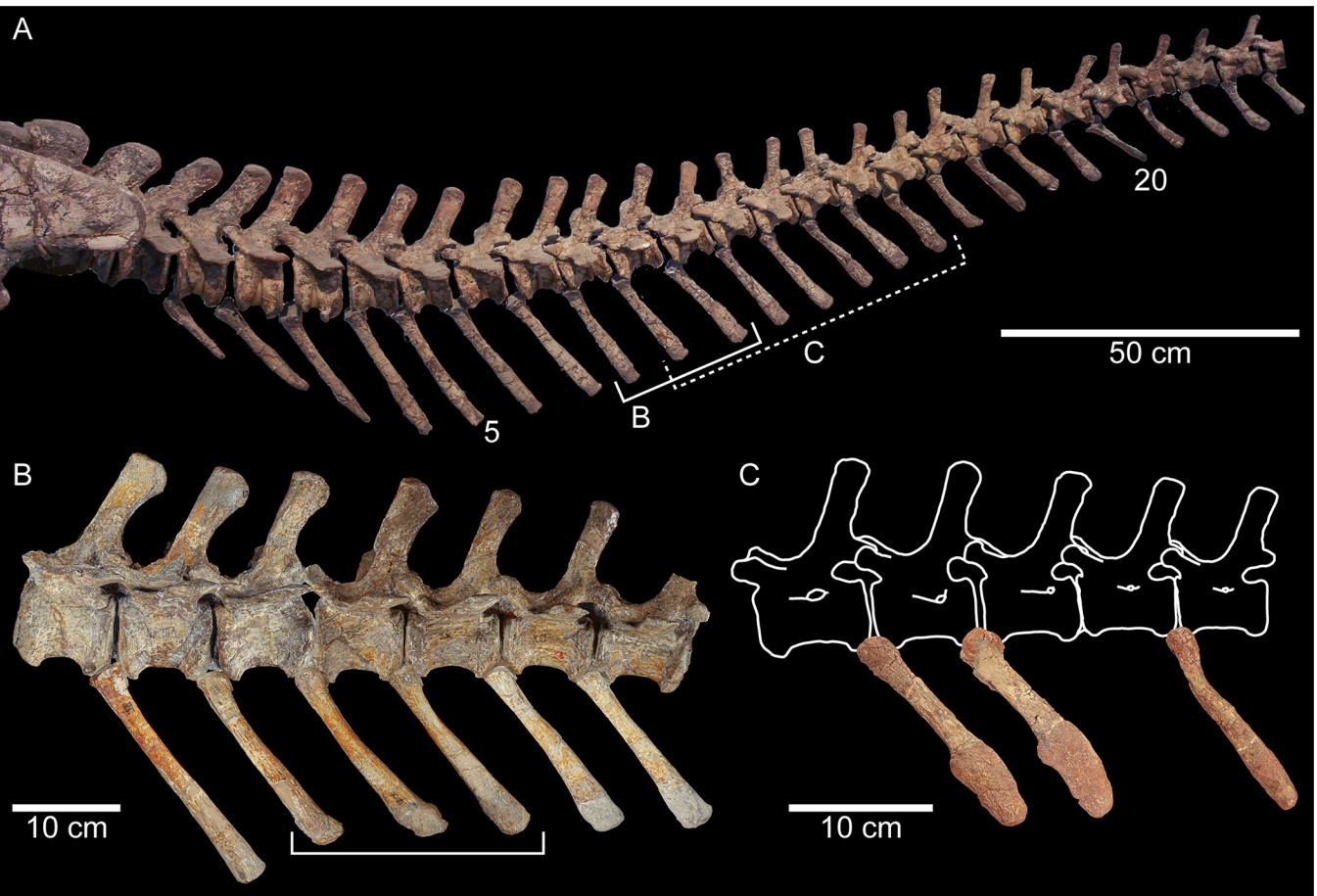

**Fig 1. Partial tails from three specimens of *Plateosaurus trossingensis* in left lateral view.** (A) First 24 caudal vertebrae and chevrons of GPIT-PV-30784 with the locations of B and C indicated using brackets; the dashed bracket for C indicates uncertainty about the position of the chevrons (see text for discussion). (B) Caudal vertebrae 7–13 and chevrons 7–12 of SMNS 13200; the bracket indicates the three pathological chevrons. (C) Pathological chevrons of SMNS 91296 with drawings of the corresponding vertebrae.

repeated periods of swelling and shrinking in a seasonal climate (for an overview of the paleoenvironmental setting, see [30]).

The deposits can be divided in three main units, Lower, Middle and Upper Beds, each representing a distinctive (yet similar between them) environmental setting and climate [30]. The two specimens herein studied come from the Lower Beds, the most fossiliferous in Trossingen, with skeletons mostly preserved in 3D [29]. The topmost two meters of the unit alone, which are accessible in Trossingen, yielded about 60 skeletons of *Plateosaurus* and three skeletons of the stem turtle *Proganochelys quenstedti*. Many of the dinosaur skeletons (usually articulated) were found in a position with their legs straight down, which has been interpreted as specimens that got mired in mud traps [27, 30, 37]. Disarticulated, yet well preserved bones are also present, which were probably transported by minor fluvial streams and/or in episodic flooding events (e.g. torrential water flows), sometimes constituting the cores of pedogenetic carbonate nodules (J.S., R.R.S., E.M., pers. obs.). In summary, there is a relatively variegated preservation of skeletal elements in the Lower Beds. The two specimens studied are generally very well preserved, especially the holotype of *P. trossingensis* (SMNS 13200), which is a virtually complete skeleton with the external surface of bones generally pristine (i.e., only minimally affected by pre-burial environmental weathering) [25]. The other specimen studied (SMNS 91296) is a

partial skeleton also relatively well preserved, though the internal structure of some bones is partially reworked and mixed with sedimentary matrix. Nevertheless, the bone structure is still recognizable.

## 3. Materials and methods

The specimens examined are housed at the Staatliches Museum für Naturkunde Stuttgart (SMNS). SMNS 13200 (Figs 1B, 2 and 3), excavated in 1912, is the holotype of *P. trossingensis* and is exceptionally well preserved and almost entirely complete [25, 30]. SMNS 91296 (Figs 1C and 4), excavated in 1932, consists of multiple individuals, clustered together by the almost 100-year-old field numbers [29]. The varying preservation, body size and the numbering

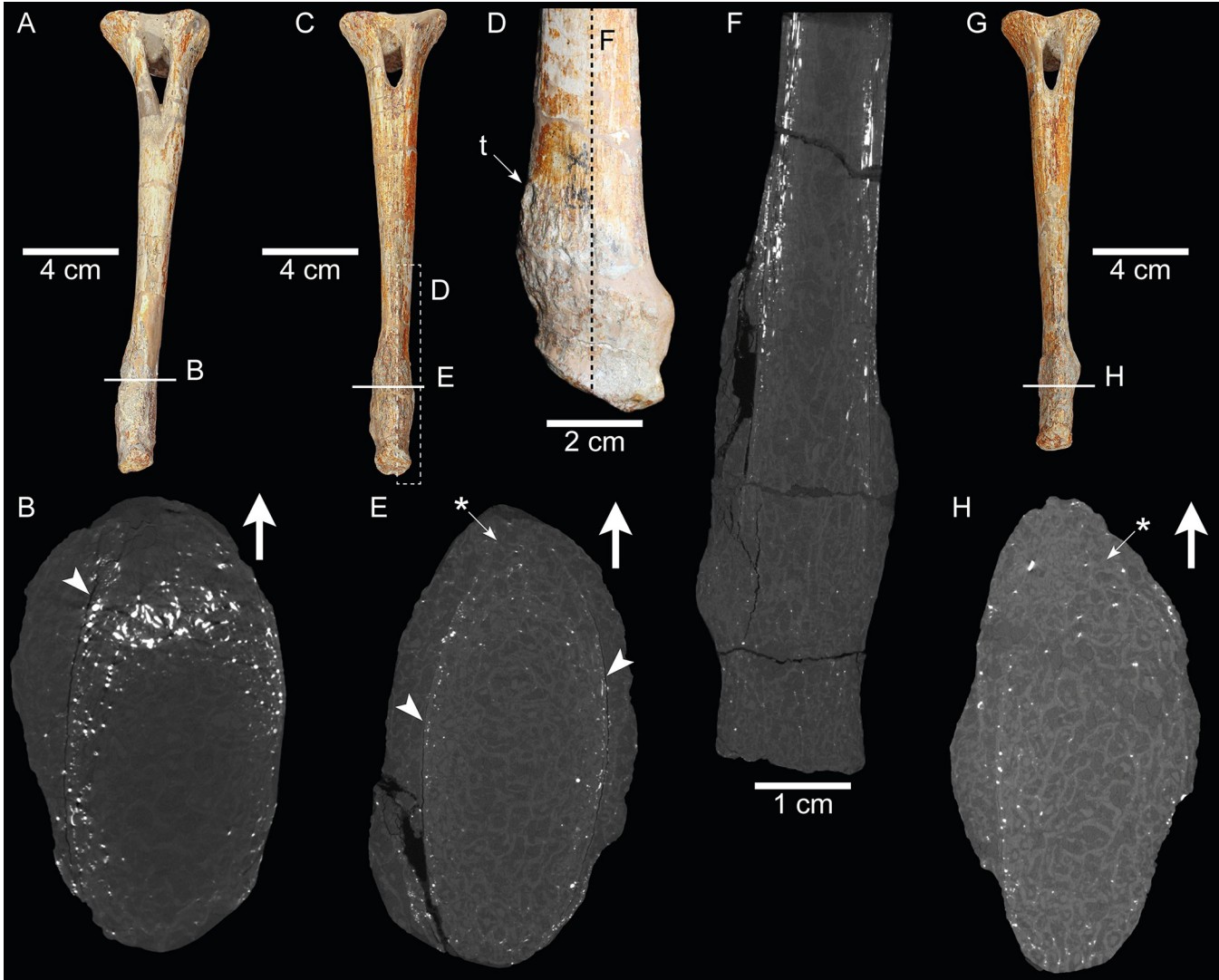

**Fig 2. Pathological chevrons of SMNS 13200.** (A) Ch8 in cranial view. (B) μCT image of Ch8 in transverse view; location of section indicated in A. (C) Ch9 in cranial view. (D) Distal end of Ch9 in left lateral view; location of view indicated in C. (E) μCT cross-section image of Ch9 in transverse view; location of section indicated in C. (F) μCT cross-section image of Ch9 in craniocaudal view; location of section indicated in D. (G) Ch10 in cranial view. (H) μCT cross-section image of Ch10 in transverse view; location of the section indicated in G. Big arrows in B, E and H indicate the cranial side of the bone. Arrowheads in B and E indicate the separation between the reactive periosteal bone and the cortical bone. "t" in D indicates a taphonomic hole. Arrows with asterisk (*) indicate the disturbance of the cortical bone on the cranial margin.

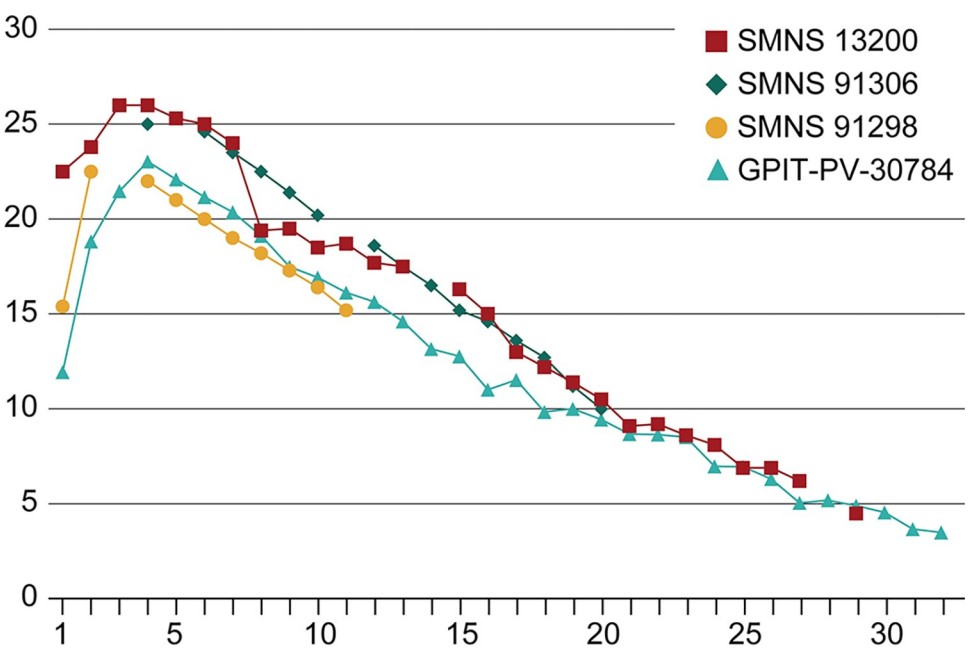

**Fig 3. Chevron lengths of four specimens of *Plateosaurus trossingensis* from the Trossingen locality.** X-axis describes the position of the chevron. Y-axis describes the length of the chevrons in centimeters. A large step in length is visible between the 7th and 8th chevrons of SMNS 13200, not present in any of the other specimens.

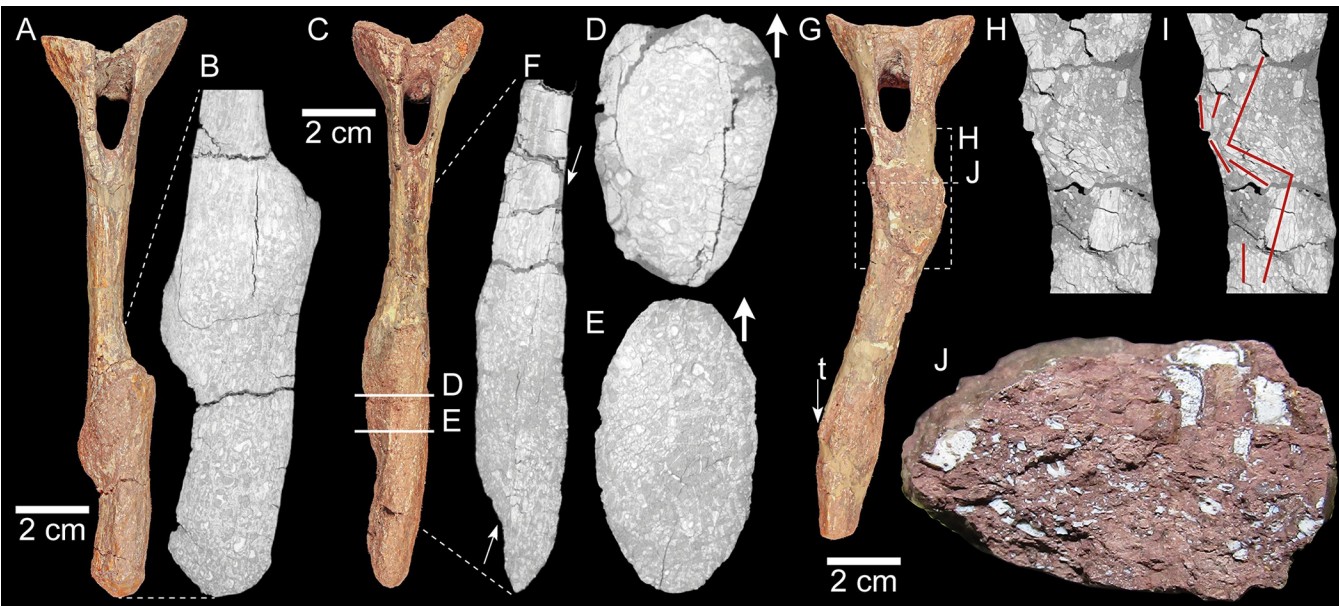

**Fig 4. Pathological chevrons of SMNS 91296.** (A) Ch"1" in cranial view. (B) μCT image of Ch"1" in craniocaudal view, representing the area indicated by the dotted line in A. (C) Ch"2" in cranial view. (D) μCT cross-section image of Ch"2" in transverse view; location of section indicated in C. (E) μCT cross-section image of Ch"2" in transverse view; location of section indicated in C. (F) μCT image of Ch"2" in craniocaudal view, representing the area indicated by the dotted line in C; arrows indicate the line separation between the original proximal and distal portions of the chevrons that were displaced. (G) Ch"3" in cranial view; "t" indicates a taphonomic fracture. (H) μCT image of Ch"3" in craniocaudal view; location of the section in G. (I) Same μCT as in H, with red lines indicating the displacement of the cortical bone. (J) Cross section of Ch"3" exposed after a recent fracture of the specimen.

present corresponding to field jackets allows the identification of 3 individuals. The individual with the pathological chevrons is a partially complete skeleton, with hindlimbs, a partial pelvis, dorsal vertebrae and forelimbs.

The specimens were μCT-scanned at the 3D Imaging Lab of the University of Tübingen with a Nikon XT H 320 μCT scanner with a 225 kV reflection target. The chevrons of SMNS 13200 were scanned at 190 kV and at 85 μA for chevron 8 (Ch8), 87 μA for Ch9 and 86 μA for Ch10. The chevrons of SMNS 91296 were scanned at 190 kV and at 90 μA for Ch"1", and 86 μA for Ch"2" and Ch"3". All specimens were scanned using a 0.1 mm copper filter, 4476 projections and two frames per projection. The μCT data were analyzed using 3D Slicer (v.5.6.0, https://www.slicer.org/). The μCT data and 3D models can be accessed on Figshare (www.figshare.com) with the DOI 10.6084/m9.figshare.26044132.

Additional pathological specimens observed are MSF 18.1, a complete specimen from Frick [38] and GPIT-PV-70385, part of a composite mount with this individual consisting of dorsal, sacral and caudal vertebrae, a partial shoulder girdle and partial front limbs, a pelvis and hindlimbs [39]. Chevron lengths were measured from Trossingen specimens with relatively complete tails: SMNS 13200, SMNS 91306, SMNS 91298 and GPIT-PV-30784 (Fig 3 and S1 Table in S1 File).

## 3.1. Institutional abbreviations

GPIT, Paleontological Collection from the University of Tübingen, Tübingen, Germany; MSF, Sauriermuseum Frick, Frick, Switzerland; PIMUZ, Paleontological Institute and Museum of the University of Zürich; SMNS, Staatliches Museum für Naturkunde Stuttgart, Stuttgart, Germany.

# 4. Results

## 4.1. Preservation and taphonomy

The pathological bones are generally well preserved, especially those of SMNS 13200 (Figs 1B and 2) [25]. However, some taphonomic modifications are also present. Ch9 displays a hole on the cranial left side ("t" in Fig 2D) that affects (cuts) both pathological and normal bone indistinctly (S1 File). This indicates that the hole was produced after the formation of the periosteal new bone. Besides this structure, the bones of SMNS 13200 seem little affected by postmortem and pre-diagenetic processes. Although also well preserved, bones of SMNS 91296 (Figs 1C and 4) appear more weathered than those of SMNS 13200. Bone surfaces are slightly cracked, resembling bones exposed for a relatively long period of time before final burial, which leads to desiccation/dehydration of the bones [40]. A long time of exposure for SMNS 91296 is also suggested by the preservation of the interior of Ch"3", visible after a recent fracture (Fig 4J): bone cavities are infilled with sedimentary matrix and part of the bone structure is removed, with bone fragments "floating" within the sediment. It should be considered that the recently fractured portion of the specimen is close to the pathological fracture of Ch"3" (Fig 4G), and thus part of the lack of structure is likely due to the traumatic event.

Besides taphonomic modifications, the historical preparation of the bones also influenced their present-day state of preservation. The relatively high abrasive techniques that were used removed part of the outer surface of the bones, as can be seen from some preparation marks that were left on the bones, as well as in the CT images (S1 File). This "overpreparation" is more marked around the pathological areas, probably because the presence of these structures was unexpected. Even with all these modifications, it is remarkable the identification of relatively small and delicate elements, such as the periosteal new bone. All in all, this shows that the locality represents indeed an exceptional fossil site, which may be considered a fossil

lagerstätte [30], especially for the concentration of elements, but likely also for the state of preservation, including abundant articulated skeletons still displaying delicate structures [25, 29].

## 4.2. Pathological chevrons in SMNS 13200

The three pathologic chevrons of SMNS 13200 are at the 8th, 9th and 10th positions in the tail (Figs 1A, 1B and 2 and S1 File) [25]. They, just like the rest of the skeleton, are very well preserved, with only minor restoration of missing parts. The chevrons have a normal dorsal end and mid-section, only the ventral end is overgrown by periosteal new bone.

In Ch8 (Fig 2A and 2B) the distal 30% of the dorsoventral length is affected by periosteal new bone formation. It creates an uneven and slightly rough surface and expands the distal end slightly in a craniocaudal and mediolateral direction. Most of the periosteal new bone is located on the right lateral margin.

In Ch9 (Fig 2C–2F) the area of periosteal new bone is more substantial than in Ch8 and ranges the distal 32% of the total length of the bone. The periosteal new bone creates an uneven and rough texture with several large prominences and relatively large pits. The periosteal new bone is relatively evenly distributed along all the sides of the chevron but forms the largest "humps" at the cranial and caudal margin.

Ch10 (Fig 2G and 2H) has the least periosteal new bone of the three as it affects the distal 29% of the bone. The majority of the periosteal new bone is located at the right and left lateral sides of the bone in the form of an ellipsoid which extends furthest laterally on the left side. Periosteal new bone is also present at the cranial margin.

The μCT images of these chevrons (Fig 2 and S1 File) show that the periosteal new bone originates from the cranial margin, where the cortex is rather indistinct and seems to be partially absent in all three specimens (asterisks in Fig 2E and 2H). The medullary cavity has a normal appearance and does not show any signs of infection or other pathologies. A gap is present between the cortex and the periosteal overgrowth (Fig 2B), indicating a probably postmortem separation between the original and the reactive bone.

Chevrons of *P. trossingensis* tend to become longer caudally between the first and the third or fourth chevron and decrease in length caudally (S1 Table in S1 File). This caudal decrease in size is relatively constant between individuals (Fig 3). The three pathological chevrons of SMNS 13200 are significantly shorter than what would be expected when compared to the surrounding chevrons: a clear, sudden decrease in length is visible between the seventh and eighth chevron, not present in any of the other skeletons (Fig 3); this is indicative of stunted growth for the injured chevrons (see discussion below).

## 4.3. Pathological chevrons in SMNS 91296

The three pathological chevrons of SMNS 91296 are most likely positioned between the 9th and the 15th vertebra (Fig 1A and 1C and S1 File). Ch"1" was most likely the most cranial one, directly followed by Ch"2". Ch"3" probably followed with one or possibly two missing chevrons in between.

Ch"1" (Fig 4A and 4B) has a clear, complete fracture at approximately 67% of the original dorsoventral length of the bone. The original length of the bone was approximately 17.7 cm. The distal end is displaced to the left dorsolateral aspect of the bone and tilted 25˚ cranially. The two halves healed together in this position and a callus formed around the overlapping area, and slightly beyond that, both proximally and distally.

In Ch"2" (Fig 4C–4F) the complete fracture is not as clear as in Ch"1". This is most likely due to overpreparation, while not realizing the presence of the pathology and how it diverges from the normal anatomy. The fracture is positioned at approximately 66% of the dorsoventral

length of the entire bone. The original length of the bone was approximately 17.6 cm. Similar to Ch"1", the distal end is displaced to the left dorsolateral aspect of the bone and is tilted 25° cranially at the distal end.

Ch"3" (Fig 4G–4J) also has a complete fracture, however it is positioned more proximally than in Ch"1" and Ch"2". The original length of the bone was approximately 15 cm, and the fracture is obliquely oriented with the center of the fracture at roughly 37% of the dorsoventral length, just ventral to the haemal canal. The oblique fracture starts caudodorsally and runs cranioventral, reaching the entire mediolateral width. The distal portion of the chevron is rotated clockwise and displaced to the right caudolateral aspect into the position the two parts are healed together. The distal end is tilted 20° caudally and 25° laterally. Close to the distal end of Ch"3" a taphonomic fracture is present, slightly offsetting the distal tip.

The μCT images of these chevrons show the displacement after fracturing (Fig 4 and S1 File). A crack in Ch"1" separating the original bone of the proximal end from the callus is observed. The distal portion is located to the side of the proximal portion, this is however not separated from the callus by a crack (Fig 4A and 4B). In Ch"2" (Fig 4C–4F), at the proximal end of the pathology there is a clear separation between the cortical bone and the surrounding overgrown callus. This is well visible in transverse view (Fig 4D), and because of the irregular line (with multiple branches) this separation is most likely because of taphonomic processes (however a perimortem fracture cannot be entirely ruled out), as in Ch"1" (Fig 4B). Slightly more distal the crack disappears and no clear cortical bone is visible (Fig 4E). Near the distal end of the lesion a thick layer of cortical bone is present with callus surrounding, not separated by a crack. In craniocaudal view a long proximodistal crack is visible separating the original dorsal portion and ventral portion of the chevron (arrows in Fig 4F). In Ch"3" the fracture occurs immediately below the haemal canal (Fig 4G). There, a clear displacement of the cortical bone with minor callus formation surrounding it is visible in mediolateral and craniocaudal view. The chevron is fractured in multiple pieces that are slightly rotated and displaced, forming a zigzagged structure (Fig 4H and 4I).

## 5. Discussion: Etiopathogenesis of lesions and paleobiological implications

In both individuals there is damage to the distal end of at least three consecutive or closely positioned chevrons, while the proximal ends and vertebrae are completely unaffected.

The three abnormal chevrons of SMNS 13200 show extensive periosteal new bone formation and a rather irregular texture on the bone surface (Fig 2). Periosteal new bone formation may be the result of several different causes–both infectious and non-infectious, and inflammatory and non-inflammatory [41]. Osteomyelitis can be ruled out since the internal structure of the bone is not affected. Pansteatitis is an inflammatory condition of the fat in crocodiles that can be severe enough to cause muscle necrosis [42] and has been reported in the tails of crocodiles. While we have a good idea of the location of skeletal muscles in the tail of *Plateosaurus* (Fig 5), we do not know the exact location of fat deposits, which we presume would differ from modern crocodilians due to differences in lifestyle and physiology. In alligators, the epaxial tail fat deposits originate medially just caudal to caudal vertebra IV and a second fat deposit begins just lateral to this deposit just caudal to caudal vertebra V (Fig 46b in [43]). The hypaxial tail fat deposits have a crescent shaped appearance as they traverse between caudofemoralis and ilio-ischiocaudalis tapering down ventrally toward the distal extent of the haemal arch/chevron (Fig 60b in [43]). With enough severity of inflammatory response, periostitis might have been possible but we would anticipate more extensive effects on the skeleton and the animal. Fat necrosis has been noted to occur in the tails of some crocodiles and reduce

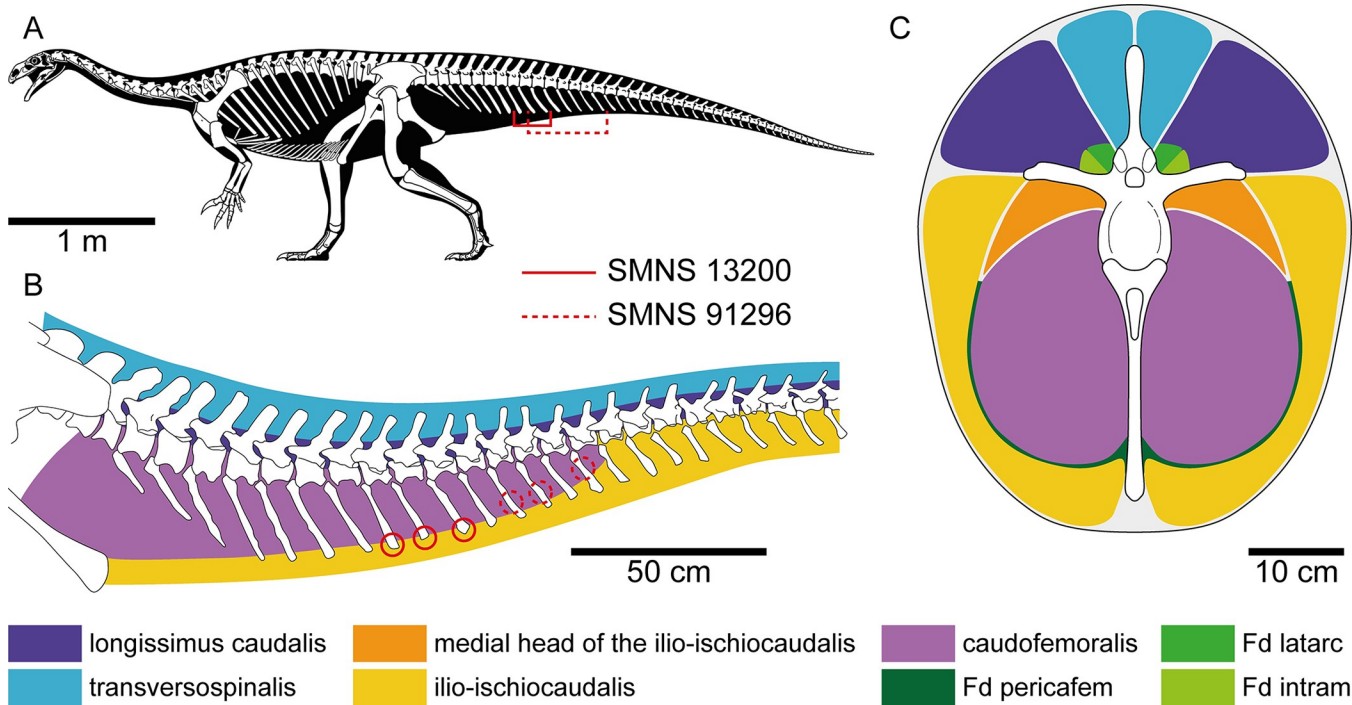

**Fig 5. Muscle reconstructions on *Plateosaurus trossingensis*.** (A) Skeletal reconstruction of GPIT-PV-30784, with the location of the pathologies on SMNS 13200 and SMNS 91296 marked (skeletal drawing and silhouette: copyright Scott Hartman, 2013; used with permission). (B) Lateral view of the anterior portion of the tail of GPIT-PV-30784 with reconstructions of the muscles in sagittal cross-section; red circles indicate the locations of the lesions for SMNS 13200 and SMNS 91296. (C) Cross-section of the tail of SMNS 13200 at the posterior end of the 6th caudal vertebra with reconstructions of muscles and fat reserves; note that the cross-section is an anatomical abstraction and depicts the neural arch and chevron in the same vertical plane. Muscle reconstructions based on Díez Díaz et al. [46], fat deposit (Fd) reconstructions based on modern crocodilians [43].

mobility by stiffening of the tails [44]. This could indirectly be related to the lesions noted as any changes in mobility might translate to abnormal stress on muscle insertions. Necrotizing fasciitis [45] has been reported to affect skeletal muscles in crocodiles and this certainly could lead to periostitis and osteomyelitis, but it is also a vicious condition that leads to mortality fairly readily with or without appropriate antibiotic treatment. On chevrons 8 to 10, the cranial side of the affected regions of the chevrons, the cortical bone appears to be missing (Fig 2E and 2H). This structure is puzzling and difficult to interpret, however, we hypothesize that it could be related to the injury. Thus, after having excluded the aforementioned possibilities, it is most probable that some form of trauma affected the chevrons of SMNS 13200 causing a tendon avulsion style injury. These injuries have not been reported in modern crocodilians to the authors' knowledge, however they have been reported in modern fowl and ratites at the gastrocnemius insertion. Given the location of the lesions, these would be at the insertion of tendons for the ilio-ischiocaudalis or caudofemoralis (Fig 5B and 5C).

A possibility that could cause these lesions is that the animal got mired in a mud trap. In the case of individuals who were mired, if the tail remained semi-fixed in place and the individual continued to thrash in an effort to break free there would have been significant forces translated to muscle insertion points. Given that multiple individuals have been found upright in a red mudstone [25, 27, 29, 30, 37], it can be inferred that miring was possible in this paleoenvironment. In modern birds, struggle when captured (an analogy for miring here) can lead to such frantic muscle strain that there can be significant muscle damage [47]. This phenomenon is known as capture myopathy. As the chevrons are extending relatively far distally from the

central axis of the tail (Figs 1A, 1B and 5A), the area can be considered vulnerable. Vulnerability has also been shown for the dorsal counterparts of the chevrons, the neural spines, which are frequently injured and in the process of healing in hadrosaurs [21, 48]. Furthermore, in the hadrosaur *Edmontosaurus*, Tanke and Rothschild [21] reported two injured chevrons displaying healed fractures.

Even though the exact nature of the trauma cannot confidently be identified in SMNS 13200, it is interesting that there is a separation between the original bone surface and the newly formed bone (arrowheads in Fig 2B and 2E). This black line observed in the CT images represents a gap that is filled with air or sediment and was formed after the death of the animal. It may indicate that there was a rather fragile connection between the new bone and the original bone surface in the living animal. Damage of the periosteum might have caused its inflammation, leading to the growth of reactive periosteal bone, originating at the cranial margin of the chevron, and also affecting the lateral sides, but for the most part leaving the caudal margin unaffected. The possibility that the three pathological chevrons represent old, healed fractures is unlikely since the μCT-scans do not show any signs of bone fracture (S1 File). The injury of the three chevrons Ch8-Ch10 led to stunted growth, possibly by damaging the cartilage, whereas the unaffected chevrons kept growing normally, resulting in the step in length shown in Figs 1B and 3. The variation in the length of the first chevron compared to the second and third between SMNS 13200, SMNS 91298 and GPIT-PV-30784 (Figs 1 and 3, S1 Table in S1 File) can be attributed to intraspecific variation [49], as there are no signs of other pathologies in any of the specimens. Since there are no other pathologies found on SMNS 13200, this appears to be an isolated incident.

In SMNS 91296 the diagnosis of fractures and displacement of the bone fragments is straightforward. Fracture is found in all three chevrons and is positioned increasingly dorsally with the more ventral position of the chevron (Figs 1C, 4 and 5B). At least one, but potentially two chevrons are missing in between Ch"2" and Ch"3" making the total fracture range 4 or 5 chevrons. The μCT data of Ch"3" show the displacement of the cortical bone overlying itself (Fig 4G–4I) indicating the force of the trauma originated from the right side and below of the tail, which is also in agreement with the displacement of the broken portions in Ch"1" and Ch"2". As no other pathologies are present in the material of SMNS 91296, this appears to be an isolated incident as in SMNS 13200. Hypotheses for this trauma include ordinary interaction with the environment (e.g., accidental self-trauma, impact trauma from a fall) or given the degree of muscle, fat and integument cover in this area (cf. Fig 5C), a bite wound with crushing force at the site that did not cause superficial trauma to the periosteum, or a penetrating wound is also very feasible (see discussion on predation below).

For damaging the periosteum or to cause fracture in the above-described chevrons in the two *Plateosaurus* individuals, a large force would be required. In SMNS 13200, periosteal new bone formation is strongest on the cranioventral side of the chevrons, similar to the position of the potentially damaged cortical bone. This might suggest that the force causing the trauma most likely came from below. The exact cause of the bone injuries in SMNS 13200 and 91296 cannot be determined, however, one might speculate that they could have been evoked by trampling by a conspecific or by another animal, by a social interaction such as fighting, playing or mating or an accidental fall. Although the hypothesis that the lesions in the tail of *Plateosaurus* can be ascribed to a predator cannot be proven in the absence of bite marks, it is certainly interesting to give a short overview on roughly coeval predators in the Central European Basin (CEB) which–judging from their size–potentially could have caused the lesions. These predators encompass two loricatan pseudosuchians from the middle Stubensandstein (Norian), the basal form *Apatosuchus* from Pfaffenhofen [50] and the *Postosuchus*-like loricatan *Teratosaurus* from Stuttgart-Heslach [51, 52], as well as the two Norian phytosaurs

*Mystriosuchus* and *Nicrosaurus* from different German localities in the CEB [53]. From the Zbąszynek Beds, southern Poland, of middle or late Norian age, indeterminate phytosaur and theropod remains have been reported [54, 55]. With its long, slender snout, however, *Mystriosuchus* was probably piscivorous [56], and it is thus rather unlikely that it attacked a large, massive animal like *Plateosaurus*. In this respect it is interesting to note that Hungerbühler [57] found evidence that a large, undetermined archosaur (?pseudosuchian) as well as a phytosaur with robust, deep snout and heterodont dentition similar to *Nicrosaurus* were scavenging on a carcass of *Plateosaurus* ("*Sellosaurus*") *gracilis*. According to this author, this indicates that at least certain phytosaurs were able to prey on large, terrestrial animals. This view was supported by the study of Drumheller et al. [11] who reported direct evidence (partially healed bite marks and a tooth embedded in a femur) that a phytosaur attacked a large, eight to nine meters long paracrocodylomorph ("rauisuchian"). Interestingly, the lesions in the tail of *Plateosaurus* described here are all on the ventral side and one could expect that a phytosaur would approach and attack the prey from the rear and below, as also suggested for other pseudosuchians [58, 59]. Last but not least, the coelophysoid theropod *Liliensternus* should be mentioned which occurred together with *Plateosaurus* in the upper Norian of Thuringia [60]. However, many of these predators have relatively short tooth exposure above the alveolar sulci, making bite wound penetration through the soft tissue in this part of the tail not a foregone conclusion. We hypothesize that either penetration to the level of the chevrons was not possible, or the mass of soft tissue could have dissipated the force of the bite. Therefore, as stated previously, force from a bite wound as a cause for fracture cannot be ruled out.

Aside from SMNS 13200 and SMNS 91296 from Trossingen, two additional individuals with comparable pathologies have been identified in various collections. MSF 18.1, a specimen from the Swiss locality of Frick (currently housed in the Paleontological Museum of the University of Zürich, PIMUZ), shows reactive bone growth at the cranial margin of the distal end of chevron 17 and potentially also in 16, 18 and 19, similar to SMNS 13200 ([38]; pers. obs. J.S. and E.M.). GPIT-PV-30785 from Trossingen has a likely pathology at the distal end of chevron 12, however the nature of the preservation and the very intense preparation have made identification of the pathology impossible. All these pathologies combined result in four individuals of *P. trossingensis* with a pathology in the mid-cranial to mid-section of the tail. The total number of specimens examined with at least a series of 10 chevrons was 27. This means that 14.8% of the specimens have a pathology in the mid-cranial to mid-section of their tail. Several pathologies are also known from the tails of sauropods [17], showing that this region of the body was relatively exposed to receive injuries (see also [24]). There are various reasons that, combined, explain this high number. First of all, the chevrons of *P. trossingensis*, but also of most other archosaurs, are long dorsoventrally and mediolaterally thin [25]. This results in a fragile element that can be damaged easily by forces much weaker than what would, for example, be necessary to damage limb bones. However, as stated earlier, these bones do not exist in isolation and would have been surrounded by muscle, fat, tendon, ligaments and integument that would have created significant cushioning for the element (Fig 5B and 5C). Secondly, the distal end of the chevrons appears to be not extremely important functionally. The proximal end of a chevron contains the haemal canal, which protects the caudal blood vessels. The distal end only functions as attachment site for muscles such as M. caudofemoralis longus and M. ilioischiocaudalis (cf. [46]; Fig 5B and 5C). The main functions of the tail as a whole in bipedal dinosaurs with long thick tails were as counterbalance to the weight at the cranial end of the body and assisting with locomotion [61–63]. A pathological distal end of several chevrons could still function as attachment site and would not by definition prohibit it from performing the main functions of the tail. This allows an animal with pathologies in the distal end of the chevron to continue living without much impact.

## 6. Conclusion

In this work we report the first pathologies in the well-known Late Triassic sauropodomorph *Plateosaurus trossingensis*. Two specimens from Trossingen, SW Germany are examined in detail, each showing a different type of pathology within the chevrons. The exact causes of the pathologies cannot be determined, however, a fitting possibility for the holotype of *P. trossingensis*, SMNS 13200, would be tendon avulsions on three chevrons, possibly caused by thrashing of the tail while being mired in one of the mud traps present at the locality it was found in. The other specimen, SMNS 91296 has pathologies that could have been caused by various reasons including ordinary trauma such as a fall or accidental self-trauma. The lesions in both specimens could also have been caused by a predator attack where the bite did not pierce the thick muscle covering of the tail all the way to leave any tooth marks on the bone.

The chevrons of *Plateosaurus* were a somewhat vulnerable part of its tail, despite the probable thick muscle covering around it. A broader overview of this taxon in multiple collections from multiple localities shows that 14.8% of the observed specimens have a pathology in the chevrons around the cranial-mid and the mid-section of the tail. We suggest that, as long as there is no severe functional damage inflicted, the animal can continue living without too many complications and the tail and accompanying chevrons will still be able to perform their main functions.

## Supporting information

**S1 File. Length in centimeters of chevrons of four specimens of *Plateosaurus trossingensis* from the Trossingen locality, and link to download of the 3D models and μCT scans.** (DOCX)

## Acknowledgments

We thank the Saurierkommision Frick and Andrea Oettl (MSF), Ben Pabst (Sauriermuseum Aathal), Daniela Schwarz (Museum für Naturkunde, Berlin), Ingmar Werneburg (GPIT) and Omar Rafael Regalado Fernández (GPIT) for the access to specimens. Torsten Scheyer (PIMUZ) and Sina Depuis (PIMUZ) are thanked for access to the unpublished Master's thesis of the latter. Isabell Rosin (SMNS) is thanked for her help in preparing the fragile material for travel to the scanning facilities. Christina Kyriakouli and the 3D Imaging Lab of the University of Tübingen are thanked for μCT scanning. We thank Scott Hartman for permission to use and providing a high resolution image of the *Plateosaurus* skeletal reconstruction with silhouette in Fig 5A. The academic editor Judith Pardo-Pérez and an anonymous reviewer are thanked for their comments which improved this manuscript.

## Author Contributions

**Conceptualization:** Joep Schaeffer, Ewan Wolff, Florian Witzmann, Rainer R. Schoch, Eudald Mujal.

**Data curation:** Joep Schaeffer, Gabriel S. Ferreira, Rainer R. Schoch, Eudald Mujal.

**Formal analysis:** Joep Schaeffer, Ewan Wolff, Florian Witzmann, Eudald Mujal.

**Funding acquisition:** Rainer R. Schoch.

**Investigation:** Joep Schaeffer, Ewan Wolff, Florian Witzmann, Eudald Mujal.

**Methodology:** Joep Schaeffer, Ewan Wolff, Florian Witzmann, Gabriel S. Ferreira, Eudald Mujal.

**Project administration:** Joep Schaeffer, Rainer R. Schoch, Eudald Mujal.

**Resources:** Gabriel S. Ferreira, Rainer R. Schoch.

**Supervision:** Eudald Mujal.

**Validation:** Joep Schaeffer, Ewan Wolff, Florian Witzmann, Gabriel S. Ferreira, Rainer R. Schoch, Eudald Mujal.

**Visualization:** Joep Schaeffer, Ewan Wolff, Rainer R. Schoch, Eudald Mujal.

**Writing – original draft:** Joep Schaeffer.

**Writing – review & editing:** Joep Schaeffer, Ewan Wolff, Florian Witzmann, Gabriel S. Ferreira, Rainer R. Schoch, Eudald Mujal.

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
