## [Decision Letter · Decision Letter 0]

10 Jun 2024

PONE-D-24-09200Paleobiological implications of chevron pathology in the sauropodomorph Plateosaurus trossingensis from the Upper Triassic of SW GermanyPLOS ONE

Dear Dr. Schaeffer,

Thank you for submitting your manuscript to PLOS ONE. After careful consideration, we feel that it has merit but does not fully meet PLOS ONE’s publication criteria as it currently stands. Therefore, we invite you to submit a revised version of the manuscript that addresses the points raised during the review process.

Please check your manuscript and answer the observations made by the reviewer, indicating the main modifications and the concern stated by the reviewer have been addressed. Additionally, remove from the text all citations that have not yet been accepted or published (i.e., reviews in process) and include number 2, 3 and 4 in figure 1. They are described in the text but not in the figure. For the introduction, I advice the authors to check Pardo-Pérez et al., 2018, 2019 and 2020 papers on paleopathologies of ichthyosaurs, specially the last one regarding tail lessions:Pardo-Pérez, J., Kear, B., Maxwell, E.E. (2020). Skeletal pathologies track body plan evolution in ichthyosaurs. Scientific Reports. 10(1), 4206

We look forward to receiving your revised manuscript.

Kind regards,

Judith Pardo-Pérez, Ph.D

Academic Editor

PLOS ONE

Journal Requirements:

2. In your manuscript, please provide additional information regarding the specimens used in your study. Ensure that you have reported human remain specimen numbers and complete repository information, including museum name and geographic location. 

For more information on PLOS ONE's requirements for paleontology and archeology research, see https://journals.plos.org/plosone/s/submission-guidelines#loc-paleontology-and-archaeology-research.

Reviewers' comments:

Reviewer's Responses to Questions

**Comments to the Author**

1. Is the manuscript technically sound, and do the data support the conclusions?

Reviewer #1: Yes

2. Has the statistical analysis been performed appropriately and rigorously? 

Reviewer #1: Yes

3. Have the authors made all data underlying the findings in their manuscript fully available?

Reviewer #1: Yes

4. Is the manuscript presented in an intelligible fashion and written in standard English?

Reviewer #1: Yes

5. Review Comments to the Author

Reviewer #1: Schaeffer et al. present a comprehensive analysis and discussion on a series of pathological chevrons in different Plateosaurus specimens from Germany and Switzerland. The authors provide a full description of the injured bones from a gross, external morphological point of view to an inner analysis via microCT scan, at this point a must-to-have when describing dinosaur pathologies. The results they propose are clear and in line with the observations made with these analyses. I do really appreciate this "population" approach, by detecting the number of pathologies among many specimens from a single locality (as already done by other authors for different species of dinosaurs), and this analysis reinforce the validity of such approach.

I have nothing to say specifically on this neat manuscript, perhaps I have only one suggestion (hence my "minor revision"): I would like to have one or two lines, at the end of the introduction (before the geological setting), to tell me which are the specific objectives of the paper. "here we focus on two individuals [...] with three consecutives chevrons displaying pathologies"... for? how? What kind of analysis will you use? What kind of information do you need to provide? What kind of implications are you expecting? The introduction seems to end abruptly, and it might cause the reader to be a bit confused (at least, this is the feeling I had while going through it). Apart from this suggested addition, I wish to congratulate the authors for the manuscript, and I will look forward to see it published.

6. PLOS authors have the option to publish the peer review history of their article (what does this mean?). If published, this will include your full peer review and any attached files.

Reviewer #1: No

---

## [Author Response · Author response to Decision Letter 0]

17 Jun 2024

Comments of the editor:

Remove from the text all citations that have not yet been accepted or published

A. We have changed the citation of “Schaeffer (in review)” to “Schaeffer 2024”, as it was published last month. The reference is added to the list. 

Include number 2, 3 and 4 in figure 1. They are described in the text but not in the figure. 

A. This was a misunderstanding. We tried to refer to figures 1, 2, 3 and 4. We have adapted the figure citations in the text to be in line with the guideline of PLoS ONE. 

For the introduction, I advice the authors to check Pardo-Pérez et al., 2018, 2019 and 2020 papers on paleopathologies of ichthyosaurs, specially the last one regarding tail lessions: Pardo-Pérez, J., Kear, B., Maxwell, E.E. (2020). Skeletal pathologies track body plan evolution in ichthyosaurs. Scientific Reports. 10(1), 4206

A. Thank you for the suggestion. We had already read these papers, and have now added Pardo-Pérez et al. 2020 to the list of used papers. 

Comments of the reviewer

I would like to have one or two lines, at the end of the introduction (before the geological setting), to tell me which are the specific objectives of the paper. "here we focus on two individuals [...] with three consecutives chevrons displaying pathologies"... for? how? What kind of analysis will you use? What kind of information do you need to provide? What kind of implications are you expecting? The introduction seems to end abruptly, and it might cause the reader to be a bit confused (at least, this is the feeling I had while going through it)

A. Thank you for these comments. We fully agree with your opinion that the introduction ends too abrupt and is lacking a proper ending. We have revised and added a few sentences at the end of the introduction to make the objectives of the study more clear. 

Additional changes

- We have changed the Morphosource link to the permanent link where the supplementary 3D models and µCT scans can be downloaded.

- We have strengthened the suggestion made in lines 402-405 of the new version of the manuscript with track changes hidden by adding a comparison to other taxa within the same group and providing two additional citations. 

- Several spelling and grammatical errors have been corrected throughout the manuscript.

---

## [Decision Letter · Decision Letter 1]

24 Jun 2024

Paleobiological implications of chevron pathology in the sauropodomorph Plateosaurus trossingensis from the Upper Triassic of SW Germany

PONE-D-24-09200R1

Dear Dr. Schaeffer,

We’re pleased to inform you that your manuscript has been judged scientifically suitable for publication and will be formally accepted for publication once it meets all outstanding technical requirements.

Kind regards,

Judith Pardo-Pérez, Ph.D

Academic Editor

PLOS ONE

Additional Editor Comments (optional):

Reviewers' comments:

Reviewer's Responses to Questions

**Comments to the Author**

1. If the authors have adequately addressed your comments raised in a previous round of review and you feel that this manuscript is now acceptable for publication, you may indicate that here to bypass the “Comments to the Author” section, enter your conflict of interest statement in the “Confidential to Editor” section, and submit your "Accept" recommendation.

Reviewer #1: All comments have been addressed

2. Is the manuscript technically sound, and do the data support the conclusions?

Reviewer #1: Yes

3. Has the statistical analysis been performed appropriately and rigorously? 

Reviewer #1: Yes

4. Have the authors made all data underlying the findings in their manuscript fully available?

Reviewer #1: Yes

5. Is the manuscript presented in an intelligible fashion and written in standard English?

Reviewer #1: Yes

6. Review Comments to the Author

Reviewer #1: The authors have properly included my only suggestion for the Introduction section, and now the "logic flow" of the paper works better.

7. PLOS authors have the option to publish the peer review history of their article (what does this mean?). If published, this will include your full peer review and any attached files.

Reviewer #1: No
